# Dietary Habits, Nutritional Knowledge, and Their Impact on Thyroid Health in Women: A Cross-Sectional Study

**DOI:** 10.3390/nu16223862

**Published:** 2024-11-12

**Authors:** Przemysław Gwizdak, Patryk Szlacheta, Daria Łaskawiec-Żuławińska, Mateusz Grajek, Karolina Krupa-Kotara, Jarosław Markowski, Ilona Korzonek-Szlacheta

**Affiliations:** 1Department of Prevention of Metabolic Diseases, Faculty of Public Health in Bytom, Medical University of Silesia, 41902 Katowice, Poland; pgwizdak@sum.edu.pl (P.G.); d201081@365.sum.edu.pl (D.Ł.-Ż.); ikorzonek@sum.edu.pl (I.K.-S.); 2Department of Basic Medical Sciences, Faculty of Public Health in Bytom, Medical University of Silesia, 41902 Katowice, Poland; pszlacheta@sum.edu.pl; 3Department of Public Health, Faculty of Public Health in Bytom, Medical University of Silesia, 41902 Katowice, Poland; 4Department of Epidemiology, Faculty of Public Health in Bytom, Medical University of Silesia, 41902 Katowice, Poland; kkrupa@sum.edu.pl; 5Department of Laryngology, Faculty of Medical Sciences in Katowice, Medical University of Silesia in Katowice, 40007 Katowice, Poland; jmarkowski@sum.edu.pl

**Keywords:** thyroid disorders, dietary habits, nutritional knowledge, women’s health, education, vitamin D, obesity, inflammatory process

## Abstract

Background: The thyroid gland plays a crucial role in regulating metabolism and various bodily functions through hormone production. Women are particularly susceptible to thyroid disorders such as hypothyroidism and Hashimoto’s disease, with associated symptoms affecting overall well-being. Prior research has inadequately addressed the influence of dietary habits and nutritional knowledge on thyroid health, especially in women. Objective: This study aimed to evaluate the dietary habits and nutritional awareness of women aged 18–45 with diagnosed thyroid disorders, emphasizing the effects of education level on knowledge and dietary practices. Material and Methods: A cross-sectional survey was conducted with 297 women diagnosed with thyroid conditions. The survey assessed demographics, comorbidities, hydration habits, and knowledge about nutrient intake critical for thyroid health. Chi-square tests, ANOVA, and correlation analyses were performed to evaluate associations. Results: Hypothyroidism and Hashimoto’s disease were most prevalent among younger women (18–25 years). A significant association was observed between higher education and knowledge of protein and carbohydrate roles in managing thyroid health (*p* < 0.01). Women with higher educational backgrounds more frequently used healthier cooking methods and were more informed about beneficial nutrients, including vitamin D and omega-3. A chi-square test indicated that low water intake was significantly associated with comorbid conditions, including insulin resistance and cardiovascular disease (*p* < 0.01). Conclusions: Significant gaps remain in dietary knowledge, particularly concerning protein intake and nutrient–drug interactions, indicating a need for targeted dietary education. Women with higher education demonstrated greater dietary awareness, emphasizing the importance of tailored educational interventions to enhance thyroid disorder management.

## 1. Introduction

The thyroid gland is a critical endocrine organ that produces key hormones, including thyroxine (T4) and triiodothyronine (T3), both of which play a pivotal role in regulating metabolism, cardiovascular and nervous system functions, and overall homeostasis. These hormones are essential in managing the metabolism of proteins, fats, and carbohydrates, while also influencing cholesterol levels and skin, hair, and nail health [1]. Disruptions in thyroid function, either through hormone overproduction or deficiency, can lead to significant metabolic imbalances, potentially resulting in conditions such as obesity, weight loss, or other metabolic disorders [2].

Women are disproportionately affected by thyroid disorders, with conditions like hypothyroidism, Hashimoto’s thyroiditis, and Graves’ disease being more prevalent in females compared to males [1]. This gender disparity is likely due to both hormonal fluctuations (such as those related to pregnancy and menopause) and immune system differences. Research has shown that autoimmune thyroid diseases, particularly Hashimoto’s thyroiditis, are more common in women, emphasizing the need for gender-specific studies to better understand how these conditions manifest and can be managed in female populations [3].

Given that thyroid dysfunction often leads to associated comorbidities—such as insulin resistance, cardiovascular disease, and hyperlipidemia—it is crucial to understand how dietary management can impact these outcomes [3,4,5]. Hypothyroidism, for example, often results in slowed metabolism and increased risk of weight gain, making appropriate nutritional support an integral part of disease management. Similarly, hyperthyroidism, which leads to accelerated metabolism, requires careful dietary adjustments to prevent complications like postprandial hyperglycemia and insulin resistance [6,7].

Despite the critical role of diet in managing thyroid health, previous research has primarily focused on general dietary recommendations without sufficiently addressing the unique needs of women with thyroid disorders [8,9]. There is a significant gap in understanding how well women are equipped to manage their conditions through nutrition and whether they possess the necessary knowledge about dietary interventions, supplementation, and hydration [10]. This gap is particularly evident in the understanding of the role of essential nutrients like selenium, zinc, and vitamin D, which have been shown to play a key role in thyroid function, especially in autoimmune thyroid diseases like Hashimoto’s [11,12,13].

Furthermore, the influence of educational background on dietary knowledge has been underexplored. Women with varying levels of education may have different access to, or understanding of, information related to thyroid health, which can affect their ability to manage their condition effectively [14]. This study aims to fill this gap by analyzing how educational level, dietary habits, and the use of dietary supplements such as vitamin D, selenium, and omega-3 fatty acids impact women with thyroid disorders [15].

A key factor in addressing thyroid health is hydration, yet its role in managing thyroid conditions remains under-examined [10,13]. Proper hydration is essential for overall metabolic function, yet many individuals, particularly women, may fail to meet recommended water intake levels, potentially exacerbating symptoms such as fatigue and constipation in thyroid disorders [1,2,3,4,5].

This study, focusing exclusively on women aged 18–45, addresses the current gap in research by evaluating the knowledge and dietary habits of this population in the context of thyroid disease management. By examining how factors such as educational background, dietary practices, and the use of supplements influence disease management, this research aims to provide a clearer picture of the unique challenges women face in managing thyroid disorders. Additionally, this study aims to uncover specific areas where targeted educational interventions can be implemented to improve knowledge and health outcomes.

By conducting a detailed analysis of how women manage their thyroid conditions through diet, hydration, and supplementation, this study contributes valuable insights into gender-specific needs in thyroid health management and underscores the importance of tailored dietary education programs.

The aim of this study is to assess dietary habits and knowledge regarding nutrition in the context of thyroid health among women with thyroid disorders, with a focus on the influence of educational attainment on these aspects.

## 2. Material and Methods

### 2.1. Study Group Characteristics

The study group consisted of 297 women, all diagnosed with thyroid disorders, and ranged in age from 18 to 45 years. Participants were recruited from GP practices and hospitals in Katowice, Poland, between March and August 2023, through a voluntary self-selection process. The participants were selected to provide a comprehensive understanding of how dietary habits and knowledge relate to the management of thyroid diseases.

Participation in this study was entirely voluntary and anonymous. Each participant provided informed consent after receiving detailed information about this study’s purpose and procedures. All respondents were assured of their privacy, with no identifying information collected, ensuring the confidentiality of the data. This ethical approach ensured that the participants felt comfortable providing accurate information about their health and dietary habits.

The diverse demographic and health profiles of the participants made it possible to explore how various factors, including age, education, and comorbid conditions, affect dietary management and knowledge in women living with thyroid disorders.

All variables were complete for all the participants, as the questionnaire required responses to all questions.

### 2.2. Research Tool

The questionnaire used in this study was specifically designed to assess the dietary habits, nutritional knowledge, and supplementation practices of women diagnosed with thyroid disorders. It consisted of 31 questions divided into two main sections: a demographic profile and a dietary assessment. The dietary section included questions related to meal frequency, types of foods consumed, hydration practices, and the use of supplements such as vitamin D, selenium, and zinc. Additionally, the questionnaire evaluated the participants’ knowledge of thyroid-friendly nutrients, cooking methods, and potential food–drug interactions.

To ensure the reliability and accuracy of the questionnaire, a comprehensive validation process was undertaken, which included both content validation and pilot testing.

In the content validation phase, a panel of five experts—including two dietitians specializing in thyroid health, an endocrinologist, and two academic researchers in nutrition and public health—reviewed the questionnaire. They assessed each item for its relevance, clarity, and comprehensiveness in relation to this study’s objectives. Based on their feedback, modifications were made to improve ambiguous questions and to add additional items that ensured the full scope of thyroid-related dietary management was covered. The relevance of each item was rated on a 1 to 4 scale, where 4 indicated high relevance. The content validity index (CVI) was calculated for each item, with a threshold of 0.80 considered acceptable. The overall CVI for the questionnaire was 0.92, indicating a high level of agreement among the experts regarding the relevance of the questions to this study.

Following content validation, the questionnaire underwent pilot testing with a small sample of 20 women diagnosed with thyroid disorders. This phase assessed the questionnaire for clarity, ease of use, and participant understanding. The participants were asked to complete the questionnaire and provide feedback on any confusing or unclear items. The pilot testing indicated that the questionnaire was well understood, with an average completion time of 12 min. Based on the feedback, minor adjustments were made to improve the wording of two questions for enhanced clarity.

To further ensure the reliability of the questionnaire, Cronbach’s alpha was calculated based on the responses from the pilot group, focusing on internal consistency. The dietary knowledge and habits section achieved a Cronbach’s alpha of 0.85, demonstrating good internal consistency. Additionally, test–retest reliability was assessed by asking the same participants to complete the questionnaire again after two weeks. The correlation coefficient for test–retest reliability was 0.88, showing that the questionnaire produced stable and consistent results over time.

In the final version of the questionnaire, all feedback from both the expert panel and the pilot group was incorporated, ensuring that the tool was comprehensive, user-friendly, and reliable. The questionnaire covered a wide range of factors important to thyroid health, including dietary intake, hydration, supplementation, and patient knowledge. Its high validity and reliability ensured accurate and consistent assessment of the dietary habits and knowledge of the women participating in this study.

Data on educational attainment and comorbidities were collected through self-reports provided by the participants within the structured questionnaire. The questionnaire included specific items asking the participants to indicate their highest level of completed education and to list any comorbid conditions they had been diagnosed with, such as insulin resistance, cardiovascular diseases, and food allergies. This approach allowed for comprehensive demographic and health profiling of the study population.

### 2.3. Ethical Considerations

This study was conducted in accordance with ethical standards for research involving human participants. Prior to participation, all the respondents were informed about this study’s objectives, procedures, and the voluntary nature of their involvement. Informed consent was obtained from each participant, and they were assured that their responses would remain anonymous and confidential. No personal identifying information was collected, ensuring the privacy of the participants.

Additionally, the participants were given the right to withdraw from this study at any time without providing a reason and without any negative consequences. This study adhered to the ethical guidelines outlined by the Declaration of Helsinki, ensuring that the rights, dignity, and well-being of the participants were prioritized throughout the research process. No interventions or manipulations were performed on the participants, as this study focused solely on gathering information through a questionnaire.

This study was conducted in accordance with the provisions of the Declaration of Helsinki and did not require the approval of the Bioethics Committee of the Silesian Medical University in Katowice (decision: PCN/CBN/0052/KB/187/22; 12 July 2022).

### 2.4. Statistical Methods

The data were compiled using Microsoft Excel 2024, and all calculations were performed using the software program Statistica 13.1. The results were analyzed to assess the knowledge and dietary habits of the participating women, with particular attention on how these habits align with current guidelines for managing thyroid conditions through diet.

A variety of statistical methods were employed to analyze the data. The chi-square test of independence was used to explore relationships between categorical variables, such as the prevalence of thyroid disorders across different age groups and the association between water intake and the presence of comorbid conditions. This test was crucial for identifying whether significant associations existed between variables by comparing observed and expected frequencies. The normality of continuous variables was verified with the Shapiro–Wilk test, and Pearson’s correlation coefficient was employed for normally distributed data.

To examine differences in dietary knowledge and practices based on educational background, an analysis of variance (ANOVA) was conducted. This method allowed for the comparison of means between different educational levels and assessed whether there were significant differences in knowledge regarding the role of protein, carbohydrate choices, and cooking methods. The ANOVA also helped determine whether educational background influenced the frequency of healthier cooking methods, such as boiling and steaming, compared to frying.

A correlation analysis was performed to examine the relationship between continuous variables, specifically meal frequency and water intake. This analysis helped identify whether an increase in one variable, such as meal frequency, was associated with a corresponding increase or decrease in water intake.

In the validation phase of the questionnaire, Cronbach’s alpha was calculated to assess the internal consistency of the dietary knowledge section, ensuring that the questions reliably measured the intended constructs. Additionally, test–retest reliability was assessed by having a pilot group complete the questionnaire twice over a period of time, allowing for the evaluation of consistency in responses across time.

For all statistical tests, the significance level was set at *p* < 0.05, ensuring that only statistically meaningful relationships were considered in the analysis.

## 3. Results

This study included 297 women aged between 18 and 45 years. The largest group of participants was aged 18–25 years (33%), followed by those aged 36–45 years (21%). Regarding educational background, the majority of the participants (64%) had obtained higher education, indicating a relatively high level of formal education. Additionally, 32% of the participants had completed secondary education, while a smaller percentage (4%) had vocational education. This diversity in educational levels provided a broad perspective on how knowledge of thyroid health and dietary management may differ across educational attainment (Table 1).

Among the women surveyed, 16% reported being diagnosed with hyperthyroidism. The highest prevalence of hyperthyroidism was observed in the 36–45 years age group (6%), while the lowest was in the 26–35 years age group (2%). The chi-square test did not show a statistically significant relationship between age and hyperthyroidism diagnosis (*p* = 0.26). Graves’ disease was reported by 11% of the women in this study, with the highest prevalence (6%) found in the women aged 45 years and older. Hypothyroidism was the most common thyroid disorder among the women surveyed, with 60% reporting a diagnosis. The 18–25 years age group had the highest prevalence (23%). A chi-square test revealed a statistically significant relationship between age group and hypothyroidism diagnosis (*p* < 0.05), indicating that younger women are more likely to be diagnosed with hypothyroidism. Similarly, for Hashimoto’s disease, the highest prevalence was in the 18–25 age group (11%), and a significant relationship was found between age group and Hashimoto’s diagnosis (*p* < 0.05). Thyroid cancer was diagnosed in 16% of the women in this study, with the highest prevalence in the 36–45 years age group (7%). However, the relationship between age group and thyroid cancer diagnosis was not statistically significant (*p* = 0.27); see Table 2.

Among the women surveyed, 54% did not report any comorbid conditions. However, 22% had food allergies or intolerances, 21% were diagnosed with insulin resistance, and 15% reported cardiovascular diseases. Other conditions, including diabetes, were present in 15% of the respondents. A chi-square test revealed a significant association between low water intake (0.5–1 L/day) and the presence of comorbidities, such as insulin resistance and cardiovascular diseases (*p* < 0.01), indicating that women with lower water intake were more likely to have these conditions.

The majority of the women (66%) relied on the internet as their primary source of information about dietary management for thyroid diseases. Other sources included books (32%), dietitian consultations (31%), and scientific publications (20%). A statistically significant association was found between the use of dietitians as a primary source of information and higher dietary knowledge (*p* < 0.01), suggesting that the women who consulted dietitians were more informed about managing their thyroid conditions.

A significant portion of the women (64%) did not have knowledge about the importance of high-quality protein in managing hypothyroidism and Hashimoto’s disease. Only 24% of the respondents indicated that increasing protein intake was important in these conditions. An ANOVA test showed a statistically significant difference in protein knowledge based on educational background (*p* < 0.01), with the women who had higher education more likely to understand the importance of protein in thyroid management. Similarly, 64% of the participants correctly identified whole-grain products as the best types of carbohydrates for individuals with thyroid diseases, while 31% were unsure. The ANOVA test also revealed a significant relationship between education and carbohydrate knowledge (*p* < 0.05).

Most of the women (76%) were aware of the beneficial role of omega-3 fatty acids in thyroid health, although 23% did not know which types of fats were the most beneficial for thyroid conditions. Vitamin D supplementation was practiced by 49% of the respondents, while only 2% used supplements like zinc and selenium. Additionally, 20% did not use any supplements. A chi-square test indicated a significant relationship between the type of thyroid disorder and vitamin D supplementation (*p* < 0.05), with the women diagnosed with Hashimoto’s disease and hypothyroidism more likely to supplement with vitamin D compared to those with hyperthyroidism or thyroid cancer.

These results are shown in Table 3.

When asked about the most beneficial vitamins and minerals for managing hypothyroidism and Hashimoto’s disease, 47% of the women identified selenium, zinc, iron, iodine, and vitamin D as key nutrients. Additionally, 82% of the respondents correctly identified fish and seafood as primary sources of iodine. Only 28% of the women were aware that vitamin C enhances the absorption of levothyroxine, a common thyroid medication. Around 46% correctly identified goitrogenic foods, such as cruciferous vegetables (e.g., cabbage, Brussels sprouts, and broccoli), which can interfere with thyroid function. However, 47% were unaware of which foods may inhibit thyroid hormone production.

More than half of the respondents (52%) reported consuming 4–5 meals per day, while only 6% consumed more than 5 meals daily. The majority of the women (90%) maintained intervals of more than two hours between meals. Only 34% of the participants consumed more than 2 L of water per day, while 14% reported a low water intake of 0.5–1 L daily. A correlation analysis found a positive relationship between meal frequency and water intake (*p* < 0.05), indicating that the women who consumed more meals per day also tended to have higher water intake.

Regarding specific food groups, 67% of the women consumed fish 1–2 times per week, and 47% consumed eggs with the same frequency. Whole-grain products were consumed regularly by 64% of the women, while 32% reported consuming legumes no more than twice a week. Vegetables were frequently consumed, with 57% eating them more than five times a week, and fruits were consumed with similar regularity, by 48% of the respondents. The most common cooking method among the women surveyed was boiling (25%), followed by steaming and frying (20%). An ANOVA test revealed a significant difference in cooking methods based on educational background (*p* < 0.01), with the women with higher education more likely to use healthier cooking methods, such as boiling and steaming, while frying was more common among those with vocational education.

In Table 4, a significant association was observed between education level and knowledge of the role of protein in thyroid health management (*p* < 0.01). It was shown that the women with higher education were more likely to know the importance of protein for thyroid health, highlighting the role of education in effective dietary management in patients with thyroid disease.

## 4. Discussion

The study results provide valuable insights into the dietary habits, hydration practices, and general knowledge of dietary management among women with thyroid diseases. These findings are essential for understanding both the strengths and gaps in patient awareness and the management of their condition, particularly through diet and lifestyle.

This study confirmed a higher prevalence of thyroid disorders among women, consistent with the findings of Vanderpump, who also indicated a higher incidence of these conditions in women [12]. Conditions such as hypothyroidism, Hashimoto’s disease, and thyroid cancer were more common in the surveyed women. The largest group of women with hypothyroidism fell within the 18–25 age range, while hyperthyroidism and Graves’ disease were more prevalent among the women aged 36–45. This age-related variation in the prevalence of different thyroid conditions highlights the need for age-specific educational and clinical interventions, which is also supported by the findings of Alevizaki et al. [16]. These findings align with the research of Carlé et al., who reported a higher incidence of thyroid disorders in women compared to men, due to hormonal and immune system differences [17].

This study revealed gaps in the participants’ knowledge of dietary management for thyroid conditions. Only 24% of the women understood the importance of increasing protein intake in the management of hypothyroidism and Hashimoto’s disease, while the majority (64%) were unaware of this key nutritional factor. This result is consistent with the research of Wiersinga et al., who emphasized the importance of protein in managing hypothyroidism [18]. Adequate protein intake is crucial for maintaining metabolic function, particularly in individuals with hypothyroidism, as protein supports muscle mass, regulates blood sugar, and maintains energy balance, which is also confirmed by Biondi and Cooper [19]. The lack of awareness on this topic suggests that many women may not be optimizing their diets for effective disease management.

A similar knowledge gap was observed regarding the types of carbohydrates beneficial for thyroid health. While 64% correctly identified whole grains as the most appropriate source, 31% of the participants were unsure. This finding aligns with Ittermann et al., who highlighted the importance of whole grains in regulating blood sugar levels in individuals with thyroid conditions [20].

Another key area was hydration practices. Only 34% of the participants consumed more than 2 L of water per day, while 14% had a daily intake of just 0.5–1 L. This result aligns with the findings of Pereira and Neves, who emphasize the importance of proper hydration in managing thyroid disorders [21]. Adequate hydration is essential for individuals with thyroid disorders, as water supports metabolic processes, aids digestion, and can alleviate symptoms of hypothyroidism, such as dry skin, fatigue, and constipation, which is also confirmed by Fenton and Silverberg [22].

Regarding general dietary habits, most participants consumed 4–5 meals per day at regular intervals of more than two hours between meals, which is consistent with dietary guidelines for stabilizing energy levels and metabolism in individuals with thyroid disorders, as also confirmed by Tan et al. [23]. However, a minority (6%) consumed more than five meals per day, which may indicate issues with overeating or insufficient portion control, in line with findings by Saponaro and Santini [24].

While 49% of the women reported taking vitamin D supplements, only 2% used essential minerals like selenium and zinc, despite their well-documented benefits in supporting thyroid function. This result is consistent with the findings of Nacamulli et al., who indicated that selenium supplementation can reduce thyroid antibody levels in patients with Hashimoto’s disease [25]. Similarly, zinc plays a crucial role in supporting the immune system and hormone production, as confirmed by Prasad [26].

Moreover, this study revealed that many of the women were unaware of how vitamin C enhances the absorption of thyroid medications; only 28% were aware of this interaction. This finding aligns with Stott et al., who emphasized the importance of proper supplementation and awareness of nutrient–drug interactions in managing thyroid diseases [27].

The most commonly used cooking methods among the participants were boiling and steaming, both suitable for preserving the nutritional integrity of foods, especially in a thyroid-friendly diet. However, a significant number of the participants (20%) still used frying, which is less ideal as it can introduce inflammatory fats that may exacerbate thyroid disorder symptoms. This result aligns with the findings of Wanjek, who noted the negative effects of frying on thyroid health [28].

This study also found that most participants relied on the internet (66%) and social media (31%) as their primary sources of dietary information, while only 20% consulted scientific publications. This finding is consistent with research by Adams and Lomax, who highlighted the risks associated with relying on online sources for health information [29]. Given the complexities of managing thyroid disorders, it is crucial for patients to access evidence-based guidance, preferably from nutrition specialists, to ensure they are following appropriate dietary and lifestyle recommendations, as supported by Mettler and Zimmermann [30].

In conclusion, this study highlights several important findings regarding the dietary habits and knowledge of women with thyroid diseases. While many participants demonstrated an understanding of basic dietary principles, significant gaps remain in their knowledge of protein intake, hydration, and the role of key supplements in thyroid health. The findings suggest a need for targeted educational interventions to improve patients’ dietary practices and awareness of nutrient–drug interactions. Additionally, there is a need to promote more reliable sources of dietary information, such as consultations with dietitians, to ensure that individuals with thyroid disorders receive accurate and beneficial guidance. This result is also consistent with research by the International Alliance of Patients’ Organizations, which underscores the importance of access to reliable health information [31].

## 5. Strengths and Limitations

This study has several strengths that contribute to its significance. First, it focuses specifically on women, a group with a higher prevalence of thyroid disorders, providing a more targeted understanding of the challenges they face in managing their condition through diet. The relatively large sample size of 97 women offers a representative view of dietary habits and knowledge gaps among this population. Additionally, the use of a custom-designed questionnaire allowed for a detailed exploration of specific dietary behaviors, hydration practices, and supplementation habits relevant to thyroid health.

However, this study also has some limitations. The reliance on self-reported data may introduce bias, as participants might not accurately recall or report their dietary intake or hydration levels. Moreover, the cross-sectional design of this study captures data at a single point in time, which limits the ability to observe changes in dietary habits or the long-term impact of educational interventions. Another limitation is the use of online platforms for distributing the questionnaire, which may have restricted participation to women with internet access, potentially excluding those from lower socioeconomic backgrounds or rural areas. Finally, this study primarily relied on descriptive statistics without more advanced analysis to determine causal relationships between dietary habits and thyroid disease management outcomes.

It should be noted that educational level may have influenced the participants’ knowledge of thyroid health, a potential confounding factor in the relationship between dietary habits and thyroid-related health outcomes. In addition, the presence of comorbidities, such as insulin resistance or cardiovascular disease, may have independently influenced the participants’ dietary choices and thyroid disease management. Consideration of these confounding factors is important for understanding the full context of the results and limitations of this study.

## 6. Conclusions

This study showed that women aged 18–25 had the highest prevalence of hypothyroidism and Hashimoto’s disease, indicating that younger women are more susceptible to these conditions. This highlights the importance of early education and intervention in managing thyroid disorders in this age group.

The women with higher education were significantly more likely to understand the role of protein and carbohydrates in managing thyroid health. This suggests a need for tailored educational efforts, particularly for the women with lower levels of education, to improve dietary management of thyroid disorders.

The women with lower water intake (0.5–1 L/day) were more likely to have comorbid conditions such as insulin resistance and cardiovascular diseases. Ensuring proper hydration is crucial for managing both thyroid health and related conditions.

The women with Hashimoto’s disease and hypothyroidism were more likely to use vitamin D supplements, but the overall use of selenium and zinc, both essential for thyroid function, remains low. Increased awareness of these nutrients is necessary for comprehensive thyroid health management.

The women who consulted dietitians had significantly higher knowledge of dietary management compared to those relying on the internet or social media. This underscores the importance of promoting professional, evidence-based dietary advice for thyroid patients.

The women with higher education were more likely to use healthier cooking methods such as boiling and steaming, while frying was more common among those with vocational education. This highlights the need for educational interventions that encourage healthier cooking practices.

Given the gaps in knowledge and dietary practices, particularly among the women with lower education, there is a clear need for targeted educational programs that address diet, hydration, and supplementation for effective thyroid management.

## Figures and Tables

**Table 1 nutrients-16-03862-t001:** Demographic characteristics of the study group (N = 297).

Demographic Characteristics	Category	N (%)
Age	18–25 years	98 (33%)
	26–35 years	65 (22%)
	36–45 years	62 (21%)
Education	Higher	190 (64%)
	Secondary	94 (32%)
	Vocational	13 (4%)

**Table 2 nutrients-16-03862-t002:** Associations between thyroid diseases and the age of the study group (N = 297).

Thyroid Conditions	Category	N (%)	Statistics	
Hypothyroidism	Overall	178 (60%)	χ^2^ = 4.12	*p* < 0.05
	Age 18–25 years	68 (23%)		
Hashimoto’s Disease	Overall	72 (24%)	χ^2^ = 5.07	*p* < 0.05
	Age 18–25 years	33 (11%)		
Thyroid Cancer	Overall	48 (16%)	χ^2^ = 3.89	*p* = 0.27
	Age 36–45 years	21 (7%)		

**Table 3 nutrients-16-03862-t003:** Associations between dietary habits and supplementation of the study group (N = 297).

Dietary Habits and Supplementation	Category	N (%)	Statistics	
Protein’s role	Awareness	71 (24%)	F = 7.45	*p* < 0.01
	Lack of awareness	190 (64%)		
Carbohydrate types	Whole-grain products	190 (64%)	F = 3.22	*p* < 0.05
	Uncertain	92 (31%)		
Vitamin D supplementation	Yes	146 (49%)	χ^2^ = 3.89	*p* < 0.05

**Table 4 nutrients-16-03862-t004:** Selected relationships demonstrated in the study (N = 297) *.

Sociomedical Data	Selected Relationship	N%	Statistics	
Age (18–25 years)	Hypothyroidism	23%	χ^2^ = 4.12	*p* < 0.05
Age (18–25 years)	Hashimoto’s disease	11%	χ^2^ = 5.07	*p* < 0.05
Education (higher)	Knowledge of the role of protein in thyroid diseases	64%	F = 7.45	*p* < 0.01
Dietitian consultation	Dietary knowledge about thyroid diseases	31%	χ^2^ = 8.21	*p* < 0.01
Hashimoto’s disease, hypothyroidism	Lack vitamin D supplementation	49%	χ^2^ = 3.89	*p* < 0.05

* χ^2^ = chi-square test result; F = ANOVA analysis result; *p* = statistical significance level.

## Data Availability

The original contributions presented in the study are included in the article, further inquiries can be directed to the corresponding author.

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
