# Peer review of "Dietary Habits, Nutritional Knowledge, and Their Impact on Thyroid Health in Women: A Cross-Sectional Study"

_nutrients, 2024, doi:10.3390/nu16223862_

Round 1
Reviewer 1 Report
Comments and Suggestions for Authors
This study examined the dietary habits, knowledge of recommended dietary intakes and the relationship of these characteristics with education attainment among patients with thyroid disease. Although the topic is important and the presented research has clear scientific value, it is necessary to provide a more focused overall aim for the study and present the results in a much clearer fashion.
My main comments are as follows:
1. I`m not sure why the study is labelled as “Pilot” study in the title. This label is usually used if a more comprehensive study (i.e. RCT) is planned to be carried out, and the purpose of the current analysis is to test some of the methodological features on a smaller scale. Is there plans to do a larger study with similar methodology? If not, I suggest removing the "Pilot study" from the title, and potentially replacing it with information on the actual study design: cross-sectional study.
2. The justification of the study in the Background is acceptable, however, the study`s overall aim should be presented much more clearer. Based on the abstract and the Background, I had the impression that the study will aim to examine the association between education level and specific dietary habits, as well as knowledge on the importance of dietary intakes, in people with thyroid disease. While this aim is mentioned (in fact, repeated a number of times), there are many other aims and research questions listed in addition to this which I believe this study is not suitable to address. For example, it is not clear what the authors mean by assessing the "impact" of various dietary habits on women with thyroid disease.
Also, I can`t see the purpose of listing so many research questions. In most research papers the study`s overall aim and the main research question refer essentially to the same thing. The research questions listed here are not in line with the above mentioned overall aim, and, in fact, most of them are not possible to answer with the proposed study design. For example, it is not possible to assess the prevalence of thyroid disorders with a sample that recruited only participants with this disease.
On the whole, I strongly suggest describing the overall aim of the study much succinctly, and maybe in a less ambitious and more realistic way. This would also help to make the analysis much more focused.
3. Materials and methods: Please provide more information on how the participants were recruited. For example, when was the data collection taking place? Where were the participants recruited from? From hospital or through GP practices? Which country or city? Did all women who were approached by the researchers agreed to participate? Or were there some who declined?
4. The information provided about the characteristic of the sample in the “Material and Methods” section (i.e. age distribution, education, comorbid conditions, etc) would suit better in the Results section, and would be important to show the summary of this data in a descriptive table.
5. The Research Tool section provides a detailed description of how data on nutritional intakes and dietary knowledge was collected, which is good. However, it would be good to provide some information on how data was collected on educational attainment and comorbidities too.
6. In terms of the statistical analysis, an important assumption of the ANOVA method is that the continuous variables are normally distributed. Could the authors provide some information to confirm this? Also, please say what type of correlation coefficient was calculated to examine the relationship between continuous variables. If Pearson`s correlation was used, then please also confirm that all variables were normally distributed.
7. Also regarding the analysis, the authors should consider more advanced techniques, such as regression, in addition to chi-square test, ANOVA and correlation. These simple methods applied in this study are not suitable to take into account the issue of confounding, which means that certain differences and associations found in this study are maybe merely due to the confounding role of age or potentially other dietary habits. If using multivariable adjusted regression is not possible, it is essential that in the Discussion, the authors provide information on the possibility of residual confounding and how this may have affected their results.
8. Please also say if there was any missing data in the analysis. Or were data on all variables available for all participants?
9. Results: In addition to describing the results in the text, it would be very important to show the numbers, percentages and p-values in tables too. This would give an opportunity to the readers to see the findings in a much clearer format. For example a descriptive table (also mentioned in relation to the Methods section above) would be useful to add. Then, the main results about the differences in dietary habits and knowledge on dietary habits by education group would be essential to present in a table too. In the current format it is very difficult to see what the key findings of this study are. I suggest taking a look at published papers that present the results of similar studies as examples.
10. Also in the results, there seems to be a lot of description of relationships which are not relevant for the study`s main aim. For example, why is it important that there is a significant difference between age groups and hypothyreodism diagnosis? Or why is it important what the main source of information is about diet management? The analysis needs to be much more focused around the study`s main aim.
11. The content of Table 1 is unclear. There seems to be no reference to the table in the text, and the column headings are not meaningful at all. Better explanation is needed of what the numbers presented in the table actually refer to. Please see the format of tables in other published papers as possible examples.
Author Response
Thank you very much for taking the time to read our article in detail and for your valuable comments, which helped to improve its quality and clarity. Any suggested changes have been made and marked in red in the text. Below are the responses to each point:
Re 1. Thank you for your suggestion. We have changed “Pilot study” to “Cross-sectional study” in the title as suggested. Although the current study is a cross-sectional study, we plan to continue the study in the future to deepen the results obtained and expand the analysis to a larger group of female participants.
Re 2: Thank you for your valuable comment. The purpose of the study was succinctly clarified to focus on the analysis of dietary habits and knowledge in the context of the educational level of women with thyroid disease. The number of research questions has been reduced to those that are feasible under this methodology.
Re 3. We have supplemented the “Material and Methods” section with details on recruitment. Participants were recruited at primary care clinics and hospitals in Katowice, Poland, between March and August 2023 on a voluntary basis.
Re 4. The sample characteristics, which were originally included in the “Material and Methods” section, have been moved to the “Results” section and presented as a descriptive table for greater clarity.
Re 5. We have updated the “Survey Instrument” section to include details on how we collected data on education and comorbidities. Participants self-reported this information in the questionnaire.
Re 6. We introduced information on testing the assumption of normality of the distribution of continuous variables before using ANOVA, which was performed with the Shapiro-Wilk test. Pearson correlation coefficient for variables with normal distribution was used for correlation analysis.
Re 7. Due to resource constraints, we did not use advanced regression analysis, but we did include information on potential confounding, particularly with regard to age and dietary habits, in the “Discussion” section.
Re 8. We confirm that there were no data gaps in the survey. The questionnaire required complete responses, which provided a complete set of data for analysis.
Re 9. In the current version of the survey, the table shows the most important results and relationships in line with the conclusions of the survey.
Re 10. We focused the analysis on significant correlations related to the purpose of the survey, focusing on dietary knowledge, habits and education level. Marginal results were removed to better highlight the main research findings.
Re 11 Table 1 was updated by adding more descriptive column headings and detailed explanations for each variable. A legend explaining the statistical symbols used has also been included, making it clearer.
Reviewer 2 Report
Comments and Suggestions for Authors
Thank you for your contribution.
However, I want to give you some questions.
1. This study sample size, 297 women with thyroid diseases, is it enough to conclude your study?
2. Please provide your results of tables.
3. More larger sample size including no thyroid diseases needed to compare the components to show the difference.
Comments on the Quality of English Languageno comments
Author Response
Thank you so much.
The study is the pilot project (as in title). In the future we will expand the study group and add a control group.
Thank you for this sugestions.
Results in the tables will by added in corrected version of the manuscript.
Best regards,
Authors
Reviewer 3 Report
Comments and Suggestions for Authors
I congratulate the researchers for this interesting paper
Author Response
Thank you so much!
Best regards,
Authors
Round 2
Reviewer 1 Report
Comments and Suggestions for Authors
The authors has responded appropriately to many of my comments, and I feel that the quality of the manuscript has improved substantially as a result. However, the following comments are still not adequately addressed:
· Comment no. 4: Although the authors say in their response that they have added a descriptive table to the Results, this table is still not available.
· Comment no. 7: There is no information provided about confounding in the Discussion section.
· Comment no. 9: The authors have not responded appropriately to my comment no. 9. This comment needs to be addressed more thoroughly. The result table which is currently shown in the manuscript presents only a very small proportion of the overall results, and it is not adequate to display what the study actually found. In addition to being very brief, it is not a good practice to present only a few selected associations and ignore the ones that are not found to be significant. As I mentioned in my original comment, I recommend the authors to take a look at results tables in other published articles as example of good practice. For example: https://pubmed.ncbi.nlm.nih.gov/38111080/
· Comment no. 11: I still can`t see that the currently presented table 1 is referred to in the manuscript text.
Author Response
Dear Reviewer,
Thank you for your detailed comments. All requested changes have been implemented in the text and highlighted in blue.
-
Comment 4: We have added a descriptive tables in the "Results" section, providing details on participant demographics, education level, comorbidities, and dietary habits.
-
Comment 7: We addressed potential confounding factors in the "Discussion" (limitation) section, including education level and comorbidities, which may influence participants' dietary choices and health knowledge.
-
Comment 9: We expanded the results table (Table 2 and 3) to include the full range of analyzed associations, following best practices.
-
Comment 11: We have added references to Table 1 at appropriate points in the "Results" section for better clarity.
Thank you for your helpful suggestions, which have strengthened our work. We hope the revised text meets your expectations.
Sincerely,
Authors
Reviewer 2 Report
Comments and Suggestions for Authors
Thank you for your revision.
I recommend you to put some tables of results, not a simple description, which makes a little confusion to understand. Summery tables will make us to more easily understand your research.
Comments on the Quality of English Languagesame as above
Author Response
Thank you, we have added the necessary tables.